# STABLE DIFFUSION FEATURE EXTRACTION FOR SKETCHING WITH ONE EXAMPLE

## ABSTRACT

Sketching is both a fundamental artistic expression and a crucial aspect of art. The significance of sketching has increased alongside the development of sketch-based generative and editing models. To enable individuals to use these sketch-based generative models effectively, personalizing sketch extraction is crucial. In response, we introduce DiffSketch, a novel method capable of generating various geometrically aligned sketches from text or images, using a single manual drawing for training the style. Our method exploits rich information available in features from a pretrained Stable Diffusion model to achieve effective domain adaptation. To further streamline the process of sketch extraction, we further refine our approach by distilling the knowledge from the trained generator into the image-to-sketch network, which is termed as DiffSketch$_{distilled}$. Through a series of comparisons, we verify that our method not only outperforms existing state-of-the-art sketch extraction methods but also surpasses diffusion-based stylization methods in the task of extracting sketches.

## 1 INTRODUCTION

Sketching, as an initial stage in artistic creation, serves as a foundational process for conceptualizing and conveying artistic intentions while visualizing the core structure and content of the final artwork. As sketches can exhibit distinct styles despite their basic form composed of simple lines, many studies in computer vision and graphics have attempted to train models for automatically extracting geometric sketches Winnemöller (2011); lllyasviel (2017); Ashtari et al. (2022); Chan et al. (2022); Seo et al. (2023). The majority of previous sketch extraction approaches utilize image-to-image translation techniques to produce high-quality results. These approaches typically require a large dataset when training an image translation model from scratch, making it difficult to personalize applications such as sketch auto-colorization, sketch-based editing, or conditional generation. Recently, advancements in abstract curve optimization have been made as an alternative that does not require training Mo et al. (2021); Vinker et al. (2022); Willett et al. (2023); Vinker et al. (2023). While these methods can effectively optimize curves based on a given text or image, they cannot follow the target style image, making it challenging to generate personalized sketches.

Meanwhile, recent research has explored the utilization of diffusion model Rombach et al. (2022); Saharia et al. (2022) features for downstream tasks Xu et al. (2023); Khani et al. (2023); Zhang et al. (2023a); Tumanyan et al. (2023). Features derived from pretrained diffusion models are known to contain rich semantics and spatial information Tumanyan et al. (2023); Xu et al. (2023), which can help train networks for various tasks using a small number of data. Previous studies have utilized these features extracted from a subset of layers Baranchuk et al. (2021), certain timesteps Zhang et al. (2023a); Xu et al. (2023), or every specific interval Luo et al. (2023). Unfortunately, these selected features often do not contain most of the information generated during the entire diffusion process.

To this end, we propose Diffsketch, a new method that can extract representative features from a pretrained Stable Diffusion (SD) Rombach et al. (2022) and train the sketch generator with one manual drawing. For feature extraction from the denoising process, we statistically analyze the features and select those that can represent the whole feature information from the denoising process. Our new generator aggregates the features from multiple timesteps, fuses them with Variational

---

The source code for both DiffSketch and DiffSketch$_{distilled}$ will be released.

Autoencoder (VAE) Kingma & Welling (2013) features, and decodes these fused features into a sketch. In addition, we distill DiffSketch into a streamlined image-to-image translation network for improved inference speed and efficient memory usage, dubbed DiffSketch$_{distilled}$.

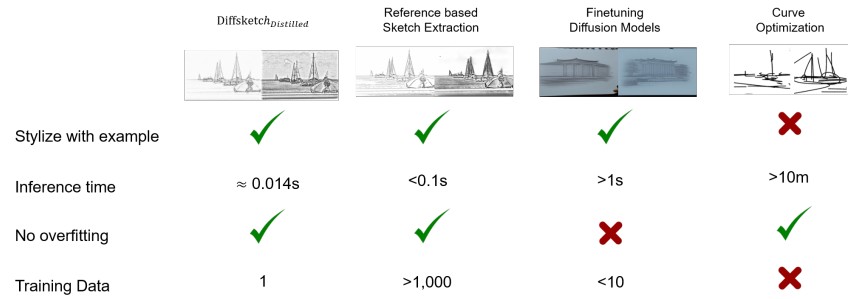

Figure 1: The uniqueness of our method: DiffSketch$_{distilled}$ is capable of extracting sketches of a given style after trained with only one example, without overfitting. DiffSketch$_{distilled}$ is different from previous methods that require large datasets Seo et al. (2023), that are prone to overfitting Ruiz et al. (2023), and that cannot extract using a style example Vinker et al. (2023).

Our method is tailored specifically for sketch generation, utilizing a dedicated sketch generator trained on features from VAE and SD. This approach sets itself apart from traditional diffusion-based personalization or stylization techniques. While existing personalization methods rely on finetuning, re-prompting, or adding adaptation modules Ruiz et al. (2023); Gal et al. (2022); Zhang et al. (2023b); Hu et al. (2021), DiffSketch employs a Decoder which is trained with a domain adaptation technique, to address common issues such as mode collapse or color leakage. It processes diffusion features without modifying the original SD model and outputs fused features in a single channel sketch. In addition, unlike curve optimization methods Vinker et al. (2022; 2023); Xing et al. (2023), DiffSketch$_{distilled}$ can control the style of sketches using a given style example. This is achieved in a few milliseconds. The differences of our method from previous methods are highlighted in Fig. 1

In addition to the newly proposed generator, we introduce a method for effective sampling performed during training. We found that training a network with data that share similar semantic information with that of the ground truth data is effective. However, relying solely on such data for training will hinder the full utilization of the capacity provided by the diffusion model. Therefore, we adopt a new sampling method to ensure training with diverse examples while enabling effective training. The resulting DiffSketch$_{distilled}$ is the final network that is capable of performing a sketch extraction task.

## 2 RELATED WORK

### 2.1 SKETCH EXTRACTION

At its core, sketch extraction utilizes edge detection. Edge detection serves as the foundation not only for sketch extraction but also for tasks like object detection and segmentation Zhang et al. (2015); Arbelaez et al. (2010). Initial edge detection studies primarily focused on identifying edges based on abrupt variations in color or brightness Canny (1986); Winnemöller (2011). Although these techniques are direct and efficient without requiring extensive datasets to train on, they often produce outputs with artifacts, like scattered dots or lines.

To make extracted sketches authentic, learning-based strategies have been introduced. These strategies excel in identifying object borders or rendering lines in distinct styles Xiang et al. (2021); Xie & Tu (2015a); lllyasviel (2017); Li et al. (2019; 2017). Informative drawing Chan et al. (2022) took a step forward from prior techniques by incorporating the depth and semantic information of images to procure superior-quality sketches. In a more recent development, Ref2sketch Ashtari et al. (2022) permits to extract stylized sketches using reference sketches through paired training. Semi-Ref2sketch Seo et al. (2023) adopted contrastive learning for semi-supervised training. All of these

methods share the same limitation; they require a large amount of sketch data for training, which is hard to gather. Due to data scarcity, training a sketch extraction model is generally challenging. To address this challenge, our method is designed to train a sketch generator using just one manual drawing.

## 2.2 DIFFUSION FEATURES FOR DOWNSTREAM TASK

Diffusion models Ho et al. (2020); Nichol & Dhariwal (2021) have shown cutting-edge results in tasks related to generating images conditioned on text prompt Rombach et al. (2022); Saharia et al. (2022); Ramesh et al. (2021). There have been attempts to analyze the features for utilization in downstream tasks such as segmentation Baranchuk et al. (2021); Xu et al. (2023); Khani et al. (2023), image editing Tumanyan et al. (2023), and finding dense semantic correspondence Luo et al. (2023); Zhang et al. (2023a); Tang et al. (2023). Most earlier studies chose a specific subset of features for their own downstream tasks. Recently, Diffusion Hyperfeature Luo et al. (2023) proposed an aggregator that learns features from all layers and that uses equally sampled time steps. We advance a step further by analyzing and selecting the features from multiple timesteps, which represent the overall features. We also propose a two-stage aggregation network and feature-fusing decoder utilizing additional information from VAE to generate finer details.

## 2.3 DEEP FEATURES FOR SKETCH EXTRACTION

Most of recent sketch extraction methods utilize the deep features of a pretrained model for sketch extraction training Ashtari et al. (2022); Seo et al. (2023); Yi et al. (2019; 2020). These approaches utilize deep features from a pretrained classifier Johnson et al. (2016); Zhang et al. (2018) or vision-language models such as CLIP Radford et al. (2021) to measure semantic similarity Chan et al. (2022); Vinker et al. (2022). They indirectly use the features by comparing them for the loss calculation during the training process instead of using them to generate a sketch. DiffSketcher Xing et al. (2023) utilizes a diffusion model to perform curve optimization from text. StyleSketch Yun et al. (2024) utilizes GAN features to extract a facial sketch with a few data. These recent models have successfully demonstrated that generative features can be used to create sketches. However, neither method can extract a sketch from a single example because DiffSketcher cannot take a style or content image as input, while StyleSketch requires 16 data for training in a single domain. To facilitate translating an image to a sketch in a provided style, we directly use the diffusion features that contain rich information and generate geometric sketches using a network trained with one example pair.

## 3 DIFFUSION FEATURES

During the backward diffusion process, UNet Ronneberger et al. (2015) produces several intermediate features with different shapes while reducing noise. This collection of features contains rich information about texture and semantics, which can be used to generate an image in various domains. For instance, features from the lower to intermediate layers of the UNet reveal global structures and semantic regions, while features from higher layers exhibit fine and high-frequency information Tumanyan et al. (2023); Luo et al. (2023). Furthermore, features become more fine-grained over time steps Hertz et al. (2022). As these features have different information depending on their embedded layers and processed timesteps, it is important to select diverse features to fully utilize the information they provide.

### 3.1 DIFFUSION FEATURES SELECTION

Here, we first present a method for selecting features by analysis. Our approach involves selecting representative features from all the denoising timesteps and building our sketch generator, $G_{sketch}$ to extract a sketch from an image by learning from a single data. To perform analysis, we randomly sampled images and collected all the features from multiple layers and timesteps during Denoising Diffusion Implicit Model (DDIM) sampling, with a total of 50 steps Song et al. (2020). For an experiment, features from a total of 50,000 data (50 UNet features with varying timesteps from 1,000 randomly generated images) were gathered.

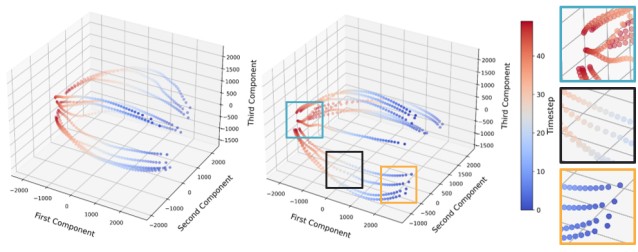

Figure 2: Analysis on sampled features. PCA is applied to DDIM sampled features, colored with denoising timesteps.

We conducted Principal component analysis (PCA) on these features from all timesteps to examine their distributions depending on their timesteps. The PCA results are visualized in Fig 2, in which smooth trajectories across the timesteps are shown. Therefore, selecting features from intervals can be more beneficial than using a single feature, as it provides richer information, as previously suggested Luo et al. (2023). Upon further examination, we can observe that the features tend to start at a similar point in their initial timesteps ($t \approx 50$) and diverge thereafter (cyan box). In addition, during the initial steps, nearby values do not show a large difference compared to those in the middle (black box), while the final features exhibit distinct values even though they are on the same trajectory (orange box).

These findings provide insights that can guide the selection of features. As we aim to capture the informative features across the timesteps instead of using all features, we first conducted a K-means clustering analysis (K-means) Hotelling (1933) using Within Clusters Sum of Squares distance (WCSS) to determine the number of feature clusters. From this process, we chose our $K$ as 13 although this $K$ value may vary with the number of diffusion sampling processes. We selected the features from the center of each cluster to use them as input to our sketch generation network. The detailed process of clustering and further experiments for different sampling processes and different models are presented in Sec. B of the Appendix.

### 3.2 Diffusion Features Aggregation

Inspired by feature aggregation networks for downstream tasks Xu et al. (2023); Luo et al. (2023), we build our two-level aggregation network and feature fusing decoder (FFD), both of which constitute our new sketch generator $G_{sketch}$. The architectures of $G_{sketch}$ and FFD are shown in Fig. 4 (b) and (d), respectively. The diffusion features $f_{l,t}$, generated on layer $l$ and timestep $t$, are passed through the representative feature gate $G^*$. They are then upsampled to a certain resolution by $U_m$ or $U_{tp}$, and passed through an aggregation network which consists of bottleneck layer ($B_l^m$ or $B_l^{tp}$) and mixing layer with mixing weights $w$. The second aggregation network receives the first fused feature $F_{fst}$ as an additional input feature.

$$F_{fst} = \sum_{t=0}^{T} \sum_{l=1}^{l_t-1} w_{l,t} \cdot B_l^m(U_m(G^*(f_{l,t}))), \; F_{fin} = \sum_{t=0}^{T} \sum_{l=l_t}^{L} w_{l,t} \cdot B_l^{tp}(U_{tp}(G^*(f_{l,t}))) + \sum_{l=l_t}^{L} w_l \cdot B_l^{tp}(U_{tp}(F_{fst}))$$

$$(1)$$

Here, $L$ is the total number of UNet layers, while $l_t$ indicates the middle layer, which are set to be 12 and 10, respectively. Bottleneck layers $B_l^m$ and $B_l^{tp}$ are shared across timesteps. $T$ is the total number of timesteps. $F_{fst}$ denotes the first level aggregated features and $F_{fin}$ denotes the final aggregated features. These two levels of aggregation allow us to utilize the features in a memory efficient manner by mixing the features sequentially in a lower resolution first and then in a higher resolution.

### 3.3 VAE Decoder Features

Unlike recent applications on utilizing diffusion features, where semantic correspondences are more important than high-frequency details, sketch generation utilizes both semantic information and high-frequency details such as texture. As shown in Fig. 3, VAE decoder features contain high-frequency details such as hair and wrinkles. From this observation, we designed our network to utilize VAE features following the aggregation of UNet features. Extended visualizations are provided in the Appendix.

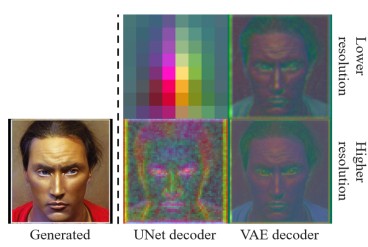

Figure 3: Visualization of features from UNet and VAE in lower and higher resolution layers. Lower resolution layers are the first layers while higher resolution layers are the 11th layer for UNet and the 9th layer for VAE.

We utilize all the VAE features from the residual blocks to build FFD. The aggregated features $F_{fin}$ and VAE features are fused together to generate the output sketch. Specifically, in the fusing step $i$, VAE features with the same resolution are passed through the channel reduction layer followed by the convolution layer. These processed features are concatenated to the previously fused feature $x_i$ and the result is passed through the fusion layer to output $x_{i+1}$. For the first step ($i = 0$), $x_0$ is $F_{fin}$. All features in the same step have the same resolution. We denote the number of total features at $i$ as $N$ without subscript for simplicity. This process is shown in Fig. 4 (d) and can be expressed as follows:

$$x_{i+1} = \text{FUSE}[\{\sum_{n=1}^{N} \text{Conv}(\text{CH}(v_{i,n}))\} + x_i], \ \hat{I}_{sketch} = \text{OUT}[\{\sum_{n=1}^{N} \text{Conv}(\text{CH}(v_{M,n}))\} + x_M + I_{source}] \tag{2}$$

where $CH$ is the channel reduction layer, Conv is the convolution layers, FUSE is the fusion layer, OUT is the final convolution layer applied before outputting $\hat{I}_{sketch}$, $\sum$ and addition represent concatenation in the channel dimension. Only at the last step ($i = M$), the source image, $I_{source}$ is also concatenated to generate the output sketch.

## 4 DIFFSKETCH

DiffSketch learns to generate a pair of image and sketch through the process described below, which is also shown in Fig. 4.

1. First, the user generates an image using a prompt with Stable Diffusion (SD) Rombach et al. (2022) and draws a corresponding sketch while its diffusion features $F$ are kept.

2. The diffusion features $F$, its corresponding image $I_{source}$, and drawn sketch $I_{sketch}$ constitute a triplet data to train the sketch generator $G_{sketch}$ with directional CLIP guidance.

3. With trained $G_{sketch}$, paired image and sketch can be generated with a condition. This becomes the input for the distilled network for fast sketch extraction.

In the following subsections, we will describe the structure of sketch generator $G_{sketch}$ (Sec. 4.1), its loss functions (Sec. 4.2), and the distilled network (Sec. 4.4).

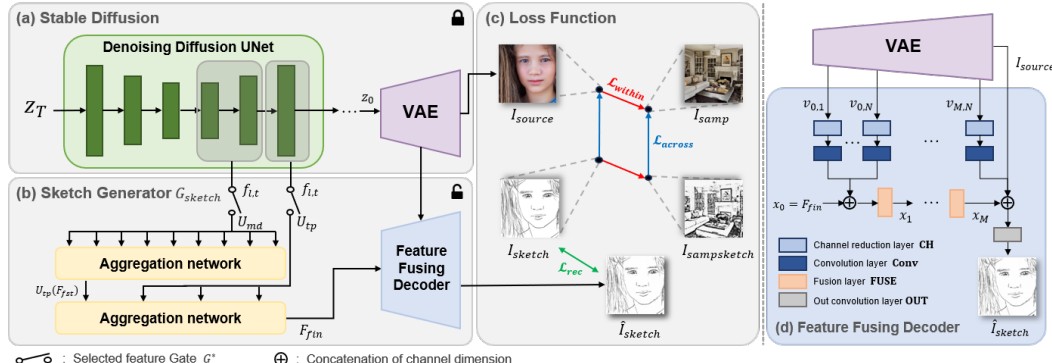

Figure 4: Overview of Diffsketch. The UNet features generated during the denoising process are fed to the Aggregation networks to be fused with the VAE features to generate a sketch corresponding to the image that Stable Diffusion generates.

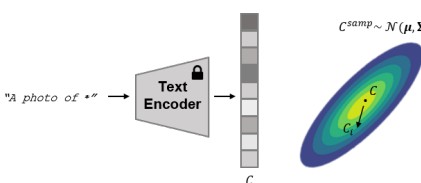

Figure 5: Illustration of CDST. Training starts with $C$, which is an encoded prompt and is diffused as the training iteration progresses to follow the distribution of SD.

### 4.1 SKETCH GENERATOR

Our sketch generator $G_{sketch}$ is built to utilize the features from the denoising diffusion process performed by UNet and VAE as described in Secs. 3.2 and 3.3. $G_{sketch}$ takes the selected features from UNet as input, and aggregate them and fuse them with the VAE decoder features $v_{i,n}$ to synthesizes the corresponding sketch $\hat{I}_{sketch}$. Unlike other image-to-image translation-based sketch extraction methods in which the network takes an image as input, our method accepts multiple deep features that have different spatial resolutions and channels.

### 4.2 OBJECTIVES

To train $G_{sketch}$, we utilize the following loss functions:

$$L = L_{\text{rec}} + \lambda_{\text{within}} L_{\text{within}} + \lambda_{\text{across}} L_{\text{across}} \tag{3}$$

where $\lambda_{\text{within}}$ and $\lambda_{\text{across}}$ are the balancing weights. $L_{\text{within}}$ and $L_{\text{across}}$ are directional CLIP losses for domain adaptation, proposed in Mind-the-gap (MTG) Zhu et al. (2022). $L_{\text{within}}$ preserves the direction within the style (image-image and sketch-sketch), by enforcing the difference between synthetic image $I_{samp}$ from SD and $I_{source}$ to be similar to that between generated sketch $I_{sampsketch}$ from $G_{sketch}$ and $I_{sketch}$ in CLIP embedding space. Similarly, $L_{\text{across}}$ enforces the difference between $I_{sampsketch}$ and $I_{samp}$ to be similar to that between $I_{source}$ and $I_{sketch}$. While MTG uses an MSE loss for the pixel-wise reconstruction, we use an L1 distance to avoid blurry sketch results, which is important in the generation of sketches Ashtari et al. (2022). Our $L_{\text{rec}}$ can be expressed as follows:

$$L_{\text{rec}} = \lambda_{\text{L1}} L_{\text{L1}} + \lambda_{\text{LPIPS}} L_{\text{LPIPS}} + \lambda_{\text{CLIPsim}} L_{\text{CLIPsim}} \tag{4}$$

where $\lambda_{\text{L1}}$, $\lambda_{\text{LPIPS}}$, and $\lambda_{\text{CLIPsim}}$ are the balancing weights. $L_{\text{L1}}$ calculates the pixel-wise reconstruction, $L_{\text{LPIPS}}$ Zhang et al. (2018) captures the perceptual similarity, and $L_{\text{CLIPsim}}$ calculates the semantic similarity in the cosine distance. More details can be found in Sec. 5.1

### 4.3 SAMPLING SCHEME FOR TRAINING

Our method uses one source image and its corresponding sketch as the only ground truth when guiding the sketch style, using the direction of CLIP embeddings. Therefore, our losses rely on well-constructed CLIP manifold. We found that when the domains of two images $I_{source}$ and $I_{samp}$ differ largely, the confidence in the directional CLIP loss becomes lower (explanation and experiment are provided in Sec. 5.2). To fully utilize the capacity of the diffusion model and produce sketches in diverse domains, however, it is important to train the model on diverse examples.

To ensure learning from diverse examples without decreasing the confidence of directional CLIP losses, we propose a novel sampling scheme, condition diffusion sampling for training (CDST) in which the condition is diffused from a single point to whole sampling space of SD. We envision that this sampling can be useful when training a model with a conditional generator. CDST initially samples a data $I_{samp}$ from one known condition encoded from prompt $C$ and gradually changes the sampling distribution to the distribution of pretrained SD by using a diffusion algorithm when training the network (see Fig. 5).

Here, to estimate the distribution of SD, we randomly sampled 100k prompts from LAION-400M Schuhmann et al. (2021) and used them as a subset of the trained text-image pairs of the SD model. We then tokenized and embedded these prompts for preprocessing, following the process of the pretrained SD model. We then conducted Shapiro-Wilk test Shapiro & Wilk (1965), followed by Mardia test Mardia (1970; 1974) with a significance level of $\alpha = 5\%$ and found that the distribution of SD follows a multivariate normal distribution. The detailed process is stated in Sec. D of the Appendix. The condition on the iteration $i$ ($0 \le i \le S$) can be described as follows:

$$\alpha_i = \sqrt{(1 - \frac{i}{S})}, \quad \beta_i = \sqrt{\frac{i}{S}}, \quad C_i^{samp} \sim \mathcal{N}(\boldsymbol{\mu}, \boldsymbol{\Sigma}), \quad C_i = \frac{\alpha_i}{\alpha_i + \beta_i} C + \frac{\beta_i}{\alpha_i + \beta_i} C_i^{samp} \tag{5}$$

where $\mathcal{N}$ represents the multivariate normal distribution, approximating the distribution of the pre-trained SD. $\mu$ represents the mean vector and $\Sigma$ represents the covariance matrix. $S$ indicates the number of the total diffusion steps during training.

## 4.4 DISTILLATION

Once the sketch generator $G_{sketch}$ is trained, DiffSketch can generate pairs of images and sketches in the trained style. This generation can be performed either randomly or with a specific condition. Due to the nature of the denoising diffusion model, in which the result is refined through the denoising process, long processing time and high memory usage are required. Moreover, when extracting sketches from images, the quality can be degraded because of the inversion process. Therefore, to perform image-to-sketch extraction efficiently while ensuring high-quality results, we train DiffSketch$_{distilled}$ using Pix2PixHD Wang et al. (2018).

To train DiffSketch$_{distilled}$, we extract 30k pairs of image and sketch samples using our trained DiffSketch, adhering to CDST. Additionally, we employ regularization to ensure that the ground truth sketch $I_{sketch}$ can be generated and discriminated effectively during the training of DiffSketch$_{distilled}$. With this trained model, images can be extracted in a given style much more quickly than with the original DiffSketch.

## 5 EXPERIMENTS

### 5.1 IMPLEMENTATION DETAILS

We implemented DiffSketch and trained generator $G_{sketch}$ on an Nvidia V100 GPU for 1,200 iterations. When training $G_{sketch}$, we applied CDST with $S$ in Eq. 5 to be 1,000. The model was trained with a fixed learning rate of 2e-4. The balancing weights $\lambda_{\text{across}}$, $\lambda_{\text{within}}$, $\lambda_{\text{L1}}$, $\lambda_{\text{LPIPS}}$, and $\lambda_{\text{CLIPsim}}$ were fixed at 1, 1, 30, 15, and 30, respectively. DiffSketch$_{distilled}$ was trained on two A6000 GPUs using the same architecture and parameters from its original paper except for the output channel, where ours was set to one. We also added regularization on every 16 iterations. DiffSketch$_{distilled}$ was trained with 30,000 pairs that were sampled from DiffSketch with CDST ($S = 30,000$). LPIPS Zhang et al. (2018), SSIM Wang et al. (2004), and FID Heusel et al. (2017) were used for a comparison with baselines while only LPIPS and SSIM were used for an ablation study due to a limited number of test data (=100). LPIPS measured perceptual similarity, SSIM measured structural similarity, and FID measured distribution similarity.

### 5.2 CONFIDENCE SCORE TEST

An underlying assumption of CDST is that for a bi-directional CLIP loss which is used for domain adaptation Yoon et al. (2024); Zhu et al. (2022); Kim et al. (2022), two images with a similar domain ($I_{source}$ and $I_{samp}$) leads to higher confidence compared to two images with a different domain. To examine this, we devised a new metric *confidence score*. As the first step, we measured similarity value $Sim_{within}$ and $Sim_{across}$ of the CLIP features of images from different domains in the same manner described as the Sec. 4.2. Specifically, the equation for similarity is as follows:

$$Sim(X, Y) = \frac{\cos(\overrightarrow{I_X I_Y} \cdot \overrightarrow{S_X S_Y}) + \cos(\overrightarrow{I_X S_X} \cdot \overrightarrow{I_Y S_Y})}{N} \tag{6}$$

where $cos(a \cdot b)$ is the cosine similarity and N is the total number for averaging. $I_X$ and $I_Y$ corresponds to the CLIP embedding of images in each domain X and Y. Similarly $S_X$ and $S_Y$ corresponds to CLIP embedding of sketches in each domain X and Y. In detail, $\cos(\overrightarrow{I_X I_Y} \cdot \overrightarrow{S_X S_Y})$ corresponds to $L_{within}$ and $\cos(\overrightarrow{I_X S_X} \cdot \overrightarrow{I_Y S_Y})$ corresponds to $L_{across}$ described in Sec. 4.2. With these computed similarities, the confidence score in domain X and domain Y can be written as follows where $Sim_{(}ALL, ALL)$ denotes the average similarity of all images, for which higher is better:

$$confidence(A,B) = \frac{Sim(X, Y)}{Sim_{(}ALL, ALL)} \times 100 \tag{7}$$

We measured the *confidence score* using 4SKST Seo et al. (2023), which consists of four different sketch styles paired with color images. 4SKST is suitable for the *confidence score* test because it contains images from two distinct domains, photos and anime, presented in four different styles. We computed a *confidence score* to determine whether the directional CLIP loss is indeed reliable when

the images for comparison are from the same domain. We conducted the test with three settings using $I_A$ (Photo) and $I_B$ (Anime), along with their corresponding sketch embeddings, $S_A$ and $S_B$. We then calculated the feature similarity within the photo domain, anime domain, and across the two domains. As shown in Table 1, for all four styles, confidence scores from the same domain were higher than those from different domains. Accordingly, we proposed a sampling scheme, CDST to train the generator in the same domain at the initial stage of the training, which leads to higher confidence while widening its capacity in the latter iterations of training.

Table 1: Confidence scores on 4SKST with four different styles.

| Similarity | Style1 | Style2 | Style3 | Style4 | Average |
|---|---|---|---|---|---|
| *confidence(Anime,Anime)* | 104.2608 | 102.8716 | 108.2026 | 101.3530 | 104.1720 |
| *confidence(Photo,Photo)* | 101.9346 | 98.8005 | 102.4516 | 100.5453 | 100.9330 |
| *confidence(Photo,Anime)* | 94.5036 | 94.0189 | 98.1867 | 92.3874 | 94.7742 |

## 5.3 DATASETS

For training, DiffSketch requires a sketch corresponding to an image generated from SD. To facilitate a numerical comparison, we established the ground truth for given images. Specifically, three distinct styles were employed for quantitative evaluation: 1) HED Xie & Tu (2015b) utilizes nested edge detection and is one of the most widely used edge detection methods. 2) XDoG Winnemöller et al. (2012) takes an algorithmic approach of using a difference of Gaussians to extract sketches. 3) Anim-informative Chan et al. (2022) employs informative learning, which is the state-of-the-art among single modal sketch extraction methods and is trained on the Anime Colorization dataset Kim (2018), which consists of 14,224 sketches. For perceptual study, we added hand-drawn sketches of two more styles. For testing, we employed the test set from the BSDS500 dataset Martin et al. (2001). As a result, our training set consisted of 3 styles and the test dataset consisted of 600 pairs (200 pairs for each style) of image-sketch for quantitative evaluation while 5 styles were used for the perceptual study. Two additional hand-drawn sketches were used only for perceptual study because there is no ground truth to compare with.

## 5.4 ABLATION STUDY

We conducted an ablation study on each component of our method compared to the baselines as shown in Table 2. Experiment were performed to verify the contribution of each component; feature selections, CDST, losses, and FFD. To perform the ablation study, we randomly sampled 100 images and extracted sketches with HED, XDog, and Anim-informative and paired them with all 100 images. All seeds were fixed to generate sketches from the same sample.

The ablation study was conducted as follows. For Random features, we randomly selected the features from denoising timesteps while keeping the number of timesteps equal to ours (13). We performed this random selection and analysis twice. For one timestep feature, we only used the features from the final timestep $t = 0$. To produce a result without CDST, we executed random text prompt guidance for the diffusion sampling process during training. For the alternative loss approach, we contrasted L1 Loss with L2 Loss for pixel-level reconstruction, as proposed in MTG. To evaluate the effect of the FFD, sketches were produced after removing the VAE features.

The evaluation results of the ablation study are shown in Table 2. Ours achieved the highest average scores for both metrics. Both Random features achieved overall low scores indicating that feature

Table 2: Quantitative results on ablation with LPIPS and SSIM. Best scores are denoted in bold.

| Sketch Styles | anim-informative | | HED | | XDoG | | **Average** | |
|---|---|---|---|---|---|---|---|---|
| Methods | LPIPS | SSIM | LPIPS | SSIM | LPIPS | SSIM | LPIPS | SSIM |
| Ours | 0.2054 | 0.6835 | **0.2117** | **0.5420** | **0.1137** | 0.6924 | **0.1769** | **0.6393** |
| Random features 1 | 0.2154 | 0.6718 | 0.2383 | 0.5137 | 0.1221 | 0.6777 | 0.1919 | 0.6211 |
| Random features 2 | 0.2042 | 0.6869 | 0.2260 | 0.5281 | 0.1194 | 0.6783 | 0.1832 | 0.6311 |
| One feature | 0.2135 | 0.6791 | 0.2251 | 0.5347 | 0.1146 | **0.6962** | 0.1844 | 0.6367 |
| W/O CDST | **0.2000** | **0.6880** | 0.2156 | 0.5341 | 0.1250 | 0.6691 | 0.1802 | 0.6304 |
| W/O L1 | 0.2993 | 0.3982 | 0.2223 | 0.5011 | 0.1203 | 0.6547 | 0.2140 | 0.5180 |
| W/O FFD | 0.2650 | 0.5044 | 0.2650 | 0.4061 | 0.2510 | 0.3795 | 0.2603 | 0.4300 |

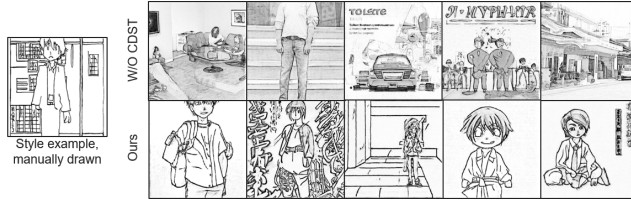

Figure 6: Comparison of results produced with and without using CDST. CDST was applied in both training and inference.

selection helps obtain rich information. Similarly, using one time step features achieved lower scores than ours on average, showing the importance of including diverse features. W/O CDST scored lower than ours on both HED and XDoG styles. W/O L1 and W/O FFD performed the worst due to lack of fine information from VAE.

**Feature Selection**   We conducted an ablation study to examine if our selected features represent all features well during the diffusion process. For this, a comparison with two baselines was made, sampling at equal time intervals (t=[i*4+1 for i in the range of (0,13)]) similar to Luo et al. (2023) and randomly selecting 13 features. We calculated the minimum Euclidean distance from each feature and confirmed that our method resulted in the minimum distance across 1,000 randomly sampled images. As illustrated in Table 3, our selected features have the lowest distance in the feature space, while selecting equally similar to Luo et al. (2023) scored the second.

Table 3: Sum of the minimum distances from all features. Our selected features better represent overall denoising features compared to sampling equally and randomly.

| Method | Ours | Equal time steps | Random sample |
|---|---|---|---|
| Euclidean Distance $(10^3)$ | 18.615 | 19.005 | 23.957 |

**Condition Diffusion Sampling for Training**   While we tested on randomly generated images (without CDST), to maintain consistency in the test set, CDST should be applied during both the training of DiffSketch and the inference for training DiffSketch$_{distilled}$. Therefore, we conducted an additional ablation study on CDST, comparing Ours (trained and sampled with CDST), against W/O CDST (trained and sampled without CDST). The outline of the sketch was clearly reproduced, following the style, when CDST was used as shown in Fig. 6.

## 5.5 COMPARISON WITH BASELINES

We initially compared our method with 11 different alternatives, including state-of-the-art sketch extraction methods Ashtari et al. (2022); Seo et al. (2023), diffusion based stylization methods Kwon & Ye (2023); Yang et al. (2023); Zhang et al. (2023c); Chung et al. (2024b), and conventional style transfer Huang & Belongie (2017). However, four of the baselines Ruiz et al. (2023); Zhang et al. (2023b); Gal et al. (2022); Chung et al. (2024a) failed or had sever artifacts thus presented in Sec. E.1 of the Appendix. Among the remaining seven baselines, Ref2sketch Ashtari et al. (2022) and S-Ref2sketch Seo et al. (2023) are methods specifically designed to extract sketches in the style of a reference by training the network on large sketch data. DiffuseIT Kwon & Ye (2023), StyleID Chung et al. (2024b), and InST Zhang et al. (2023c) are designed for diffusion based image-to-image translation by disentangling style and content. AdaIN Huang & Belongie (2017) is conventional style transfer method, and ZeCon Yang et al. (2023) is text based stylization method.

Table 4 presents the result of quantitative evaluation. Overall, ours achieved the best scores. While S-Ref2sketch scored the second highest, it relied on a large sketch dataset to train unlike ours that required only one training data. Fig. 7 presents visual results produced by different methods. While S-Ref2sketch, Ref2sketch, StyleID, and AdaIn generated comparable quality in one or two sources, they did not faithfully follow the exact style in others. DiffuseIT sometimes failed to disentangle style and content, while InST and ZeCon failed to extract sketches following the target style. DiffSketch$_{distilled}$ generated superior results compared to these baselines, effectively maintaining its styles and content.

## 5.6 PERCEPTUAL STUDY

We conducted a user study to evaluate different sketch extraction methods on human perception. We recruited 21 participants to complete a survey that used test images from five different styles, to extract sketches. Each participant was presented with a total of 20 sets of source image, target

Figure 7: Qualitative comparison with baselines.

Table 4: Quantitative comparison of different methods on the BSDS500 datasets. Best scores are denoted in bold, and the second-best are underlined.

| Methods | BSDS500 - anime | | | BSDS500 - HED | | | BSDS500 - XDoG | | | BSDS500 - average | | |
|---|---|---|---|---|---|---|---|---|---|---|---|---|
| | LPIPS | SSIM | FID | LPIPS | SSIM | FID | LPIPS | SSIM | FID | LPIPS | SSIM | FID |
| Ours | **0.218** | 0.493 | 126.5 | **0.227** | **0.593** | **110.6** | **0.143** | **0.649** | **62.8** | **0.196** | **0.578** | **100.0** |
| Ref2sketch | 0.336 | 0.469 | 155.2 | 0.420 | 0.315 | 168.6 | 0.571 | 0.131 | 274.5 | 0.442 | 0.305 | 199.4 |
| S-Ref2sketch | 0.239 | **0.510** | **99.1** | 0.397 | 0.342 | 162.3 | 0.505 | 0.309 | 192.6 | 0.380 | 0.387 | 151.3 |
| DiffuseIT | 0.484 | 0.298 | 215.2 | 0.492 | 0.191 | 214.2 | 0.573 | 0.110 | 215.3 | 0.516 | 0.200 | 214.9 |
| StyleID | 0.375 | 0.314 | 211.8 | 0.405 | 0.121 | 198.5 | 0.241 | 0.459 | 135.4 | 0.340 | 0.298 | 181.9 |
| AdaIN | 0.348 | 0.411 | 205.2 | 0.392 | 0.256 | 200.1 | 0.406 | 0.249 | 187.7 | 0.382 | 0.305 | 197.7 |
| InST | 0.677 | 0.180 | 245.9 | 0.592 | 0.129 | 187.8 | 0.477 | 0.294 | 244.3 | 0.582 | 0.201 | 226.0 |
| ZeCon | 0.702 | 0.243 | 254.6 | 0.619 | 0.160 | 253.3 | 0.494 | 0.341 | 262.5 | 0.605 | 0.248 | 256.8 |

sketch style, and resulting sketch. Participants were asked to choose one that best follows the given style while preserving the content of the source image. As shown in Table 5, our method received the highest scores among all competing methods. Ours outperformed the diffusion-based methods and even received a higher preference rating than the specialized sketch extraction method that was trained on a large sketch dataset.

Table 5: Results from the perceptual study performed given a style example and the source image. The percentages indicate the selection frequency. Ours was the most frequently chosen, with more than double the selection rate of the second-highest.

| Ours | Ref2sketch | S-Ref2sketch | DiffuseIT | StyleID | AdaIn | InST | ZeCon |
|---|---|---|---|---|---|---|---|
| **49.52%** | 1.90% | 17.38% | 1.19% | 15.48% | 8.10% | 6.43% | 0.0% |

## 6 LIMITATION AND CONCLUSION

We proposed DiffSketch, a novel method to extract sketches in given styles by training a sketch generator using representative features. For the first time, we conducted the task of extracting sketches from the features of a diffusion model and demonstrated that our method outperforms previous state-of-the-art methods. The ability to extract sketches in input style, trained with one example, will have various use cases not only for artistic purposes but also for personalizing sketch-to-image retrieval and sketch-based image editing.

We built our generator network specialized for producing sketches by fusing aggregated features with the features from a VAE decoder. Consequently, our method works well with diverse sketches including dense sketches and outlines. However, because our method utilizes features during generation, it requires the user to draw a sketch, making it impossible to use existing sketch pairs. One possible future research direction could involve utilizing features from inversion. To help understand future research in this direction, we visualize the features from inversion to show that their characteristics are similar to the features from generation in Sec. B.2 of the Appendix.

Although we focused on sketch extraction, our analysis of selecting representative features and the proposed training scheme are not limited to the domain of sketches. Extracting representative features holds potential to improve applications leveraging diffusion features, including semantic segmentation, visual correspondence, and depth estimation. We believe that this research direction promises to broaden the impact and utility of diffusion feature-based applications.

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

# APPENDIX

This appendix consists of 5 Sections. Sec. A describes implementation details. Sec. B provides additional details and findings on diffusion features selection. Sec. C presents extended details of VAE decoder features. Sec. D contains the results of additional experiments on CDST. Sec. E presents additional comparison with baselines and additional qualitative results with various style sketches.

## A. IMPLEMENTATION DETAILS

**DiffSketch**    DiffSketch leverages Stable Diffusion v1.4 sampled with DDIM Song et al. (2020) pretrained with the LAION-5B Schuhmann et al. (2022) dataset, which produced images of resolution $512 \times 512$. With the pretrained Stable Diffusion, we use a total of 50 time steps T for sampling. The training of DiffSketch was performed for 1200 iterations which required less than 3 hours on an Nvidia V100 GPU. For the training using HED Xie & Tu (2015b), we concatenated the first two layers with the first three layers to stylize sketch. In case of XDoG Winnemöller (2011), we used the Gary Grossi style.

**DiffSketch$_{distilled}$**    DiffSketch$_{distilled}$ was developed to conduct sketch extraction efficiently with the streamlined generator. The training of DiffSketch$_{distilled}$ was performed for 10 epochs for 30,000 sketch-image pairs generated from DiffSKetch, following CDST. The training of DiffSketch$_{distilled}$ required approximately 5 hours on two Nvidia A6000 GPUs. The average inference time of both DiffSketch and DiffSketch$_{distilled}$ was 4.74 seconds and 0.0139 seconds, respectively, when tested on an Nvidia A5000 GPU with 1,000 images with resolutions of 512 x 512 using automatic precision.

## B. DIFFUSION FEATURES SELECTION

### B.1 DETAILS OF DIFFUSION FEATURE SELECTION PROCESS AND ANALYSIS

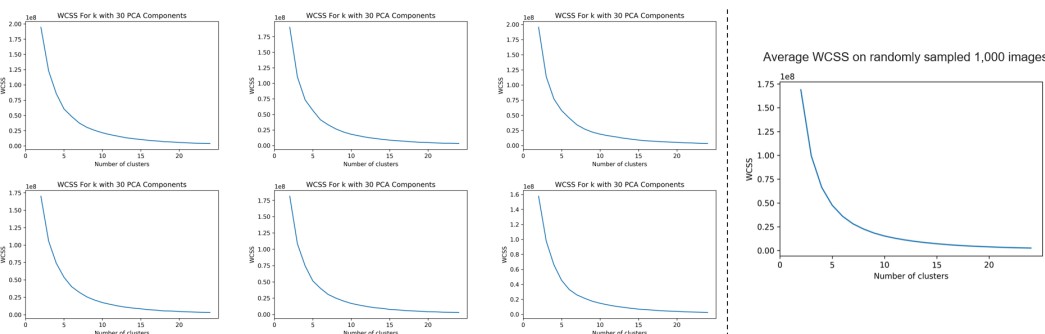

Figure 8: Visualization of WCSS values according to the number used for K-means clustering. The left plots are the WCSS values of the features from randomly sampled images while the right plot shows the average WCSS values of the features from all images.

To conduct K-means clustering for diffusion feature selection, we first employed the elbow method, visualizing the results. However, a distinct elbow was not visually apparent, as shown in Fig. 8. The left 6 images are WCSS values from randomly selected images. All 6 plots show similar patterns, making it hard to select a definitive elbow as stated in the main paper. The right image, which exhibits similar results, shows the average of WCSS on all 50,000 UNet features from 1,000 different images.

Therefore, we chose to use the Silhouette score Rousseeuw (1987) and Davies-Bouldin index Davies & Bouldin (1979), which are two of the most widely used numerical methods when choosing the optimal number of clusters. However, they are two different methods, whose results do not always match with each other. We first visualized and found the contradicting results of these two methods

as shown in Fig. 9. Therefore, we chose to use the one that first matches the $i^{\text{th}}$ highest silhouette score and the $i^{\text{th}}$ lowest Davies-Bouldin index simultaneously. This process of choosing the optimal number of clusters can be written as follows :

---

**Algorithm 1** Finding the Optimal Number of Clusters

1: $MAX\_clusters = Total\_time\_steps/2$
2: $sil\_indicies \leftarrow$ sorted(range $(MAX\_clusters)$, key $= \lambda k : silhouette\_scores[k],$ reverse $= True$)
3: $db\_indicies \leftarrow$ sorted(range $(MAX\_clusters)$, key $= \lambda k : db\_scores[k],$ reverse $= False$)
4: **for** $i \leftarrow 0$ **to** $MAX\_clusters$ **do**
5:     **if** $sil\_indicies[i]$ **in** $db\_indicies[: i + 1]$ **then**
6:         $k\_optimal = sil\_indicies[i]$+1
7:         **break**
8:     **end if**
9: **end for**

---

We conducted this process twice with two different numbers of PCA components (10 and 30), yielding the results shown in Fig. 10. The averages (13.26 and 13.34) and standard deviations (0.69 and 0.69) were calculated. As the mode value with both PCA components was 13, and the rounded average was also 13, we chose our optimal k to be 13. Using this number of clusters, we chose the representative feature as the one nearest to the center of each cluster. From this process, we ended up with the following t values: [0, 3, 8, 12, 16, 21, 25, 28, 32, 35, 39, 43, 47].

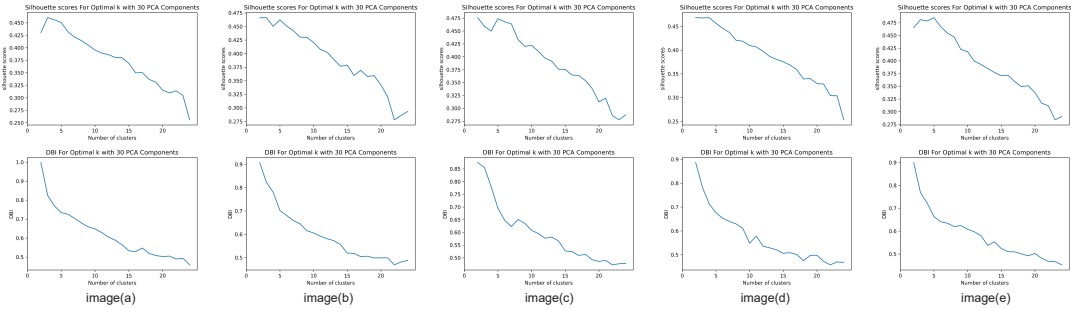

image(a)      image(b)      image(c)      image(d)      image(e)

Figure 9: Visualization of contradicting results of Silhouette scores and Davis Bouldin indices on five different images.

In the main paper, we identified several key insights through the visualization of features. For future research and to provide additional insights, we manually classified images and visualized the trajectory of features from different classes as shown in Fig. 11. Here, we summarize extensively about our findings through feature analysis. First, semantically similar images lead to similar trajectories, although not identical. Second, features in the initial stage of the diffusion process (when t is approximately 50) retain similar information despite significant differences in the resulting images. Third, features in the middle stage of the diffusion process (when t is around 25) exhibit larger differences between adjacent features in their time steps. Lastly, the feature at the final time step (t=0) possesses distinctive information, varying significantly from previous values. This is also evident in the additional visualization presented in Fig. 11.

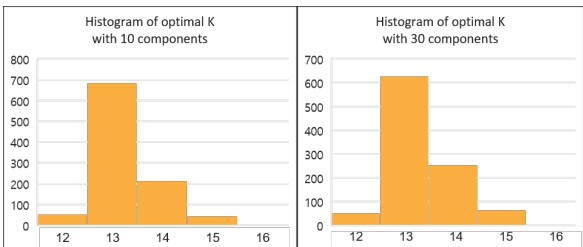

Figure 10: Visualization of histogram for the optimal k value with different numbers of PCA components.

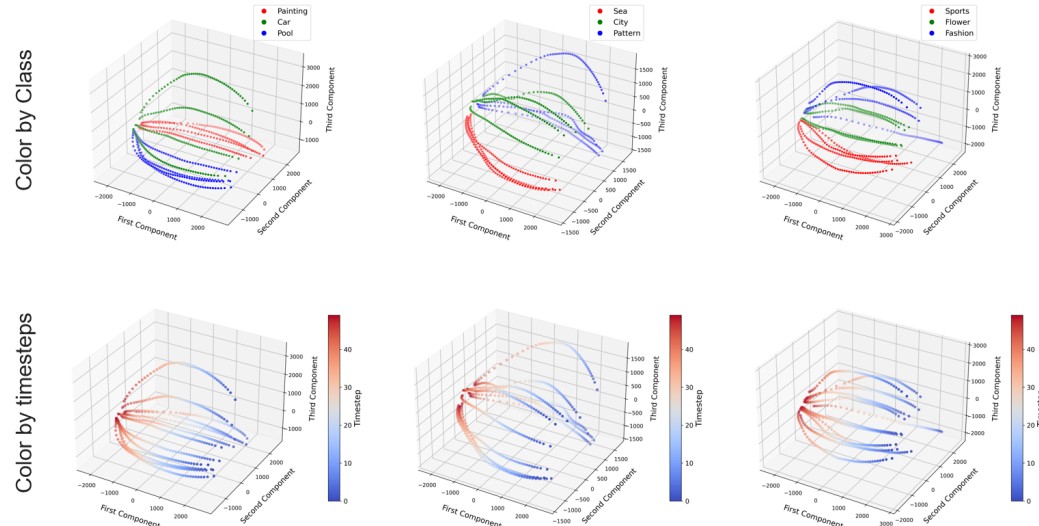

Figure 11: Additional analysis on sampled features. PCA is applied to DDIM sampled features from different classes. Up : features colored with human-labeled classes. Down : features colored with denoising timesteps

The automatically selected features indicate a prioritization of the final feature (t=0), and the selection was made more from middle steps than from initial steps (t=[21,25,28] versus t=[43,47]). Our finding offers some guidance for manual feature selection to consider the time steps, especially when memory is constrained. The order of the preference is the features from the last step (t=0), from the middle (t is near 25), and from middle to final time steps while the features from initial steps are preferred less in general. For instance, when selecting five features from 50 time steps, a possible selection could be t=[0, 10, 20, 30, 40] instead of using simple equal timesteps (t = [9,18,27,36,45]). However, for a task of semantic correspondence or segmentation, it is known that features from last 0 to 30% are more informative Baranchuk et al. (2021); Xu et al. (2023); Tang et al. (2023); Zhang et al. (2023a), therefore one possible choice can be [0, 7, 14, 24, 34].

## B.2 FEATURES FROM INVERSION, DIFFERENT STEPS, AND MODEL

While we focused on T=50 DDIM sampling, for generalization, we examined different intervals (T=25, T=100) and different models. For these experiments, we randomly sampled 100 images. While our previous experiments reported in Fig. 11 were conducted with manually classified images, we utilized DINOv2 Oquab et al. (2023), which was trained in a self-supervised manner and has learned visual semantics. With DINOv2, we separated the data into 15 different clusters and followed the process described in the main paper to plot the features. Here, we used 15 images from each cluster to calculate the PCA axis while we used 17 classes in the main experiments. The results, as shown in Fig. 13 and Fig. 14, indicate that even with different sampling methods, the same conclusions regarding the sampling method can be drawn. The last feature exhibits a distinct value, while the features from the initial time step have similar values.

In addition, we also tested on features extracted during the inversion process. We randomly selected 20 images from human face Karras et al. (2019) and cat photos Choi et al. (2020) to plot the features as shown in Fig. 12. Lastly, we tested on another model, Stable diffusion V2.1 which produces 768×768 images. Following the same process, we randomly sampled 100 images and clustered with DINOv2 and plot the results as shown in Fig. 15. This result also shows that even with different models with different resolutions, the same conclusions can be drawn, showing the scalability of our analysis.

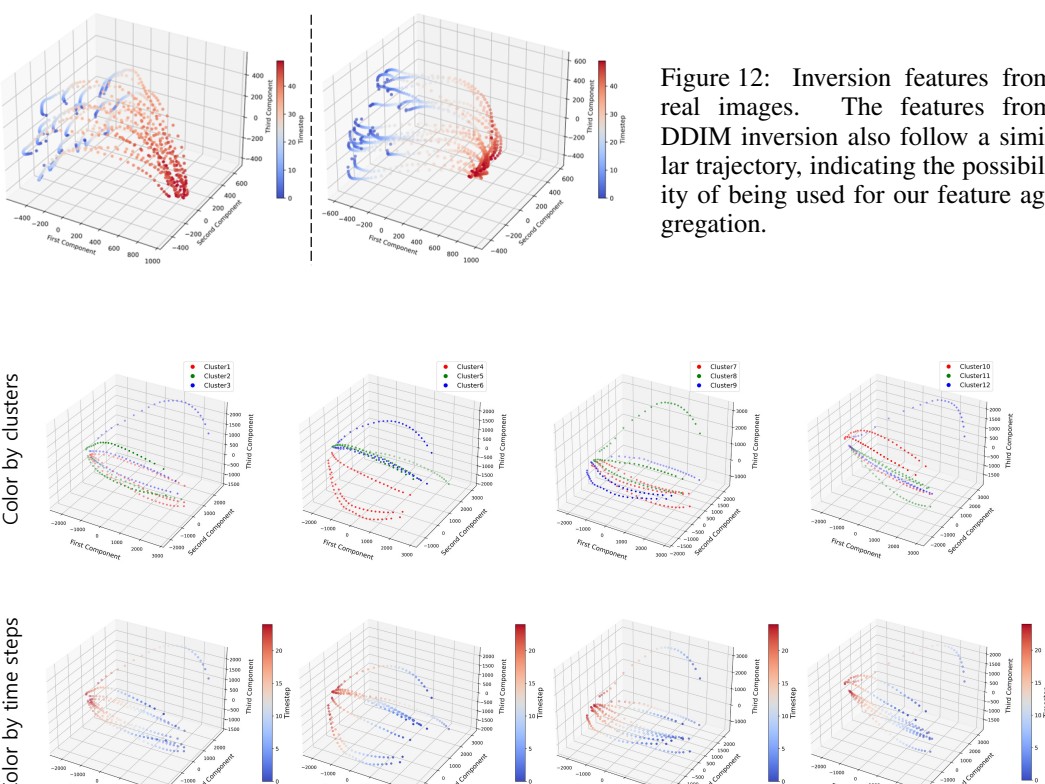

Figure 12: Inversion features from real images. The features from DDIM inversion also follow a similar trajectory, indicating the possibility of being used for our feature aggregation.

Figure 13: Additional analysis on sampled features. PCA is applied to 25 steps of DDIM sampled features with different clusters. Up : features colored with DINOv2 clusters. Down : features colored with denoising timesteps.

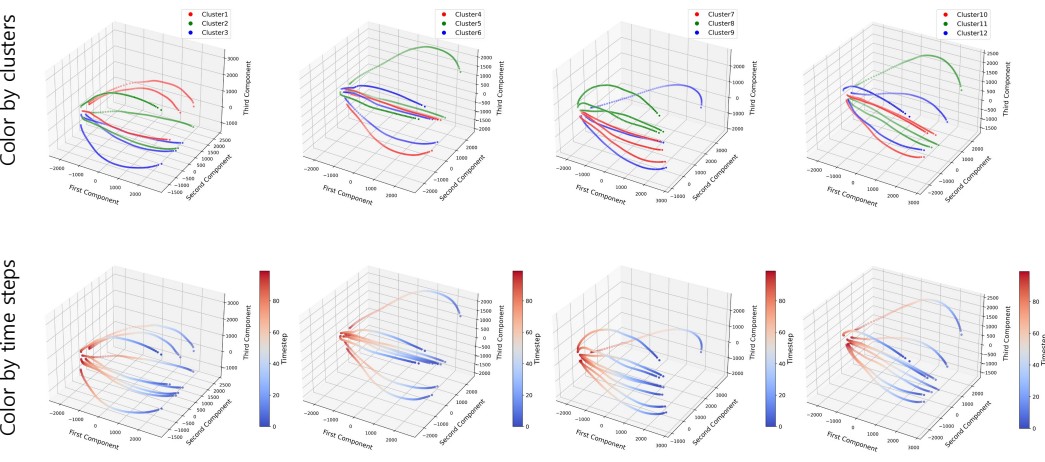

Figure 14: Additional analysis on sampled features. PCA is applied to 100 steps of DDIM sampled features with different clusters. Up : features colored with DINOv2 clusters. Down : features colored with denoising timesteps.

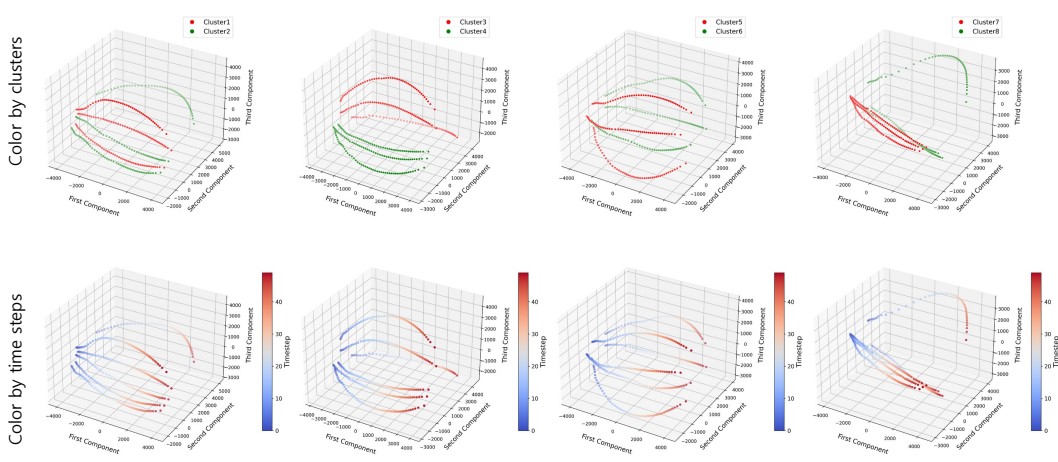

Figure 15: Additional analysis on Stable diffusion v2.1 sampled features. PCA is applied to 50 steps of DDIM sampled features with different clusters. Up : features colored with DINOv2 clusters. Down : features colored with denoising timesteps.

## C. VAE DECODER FEATURES

The VAE features were fused with the Aggregation network features using FFD in the proposed model architecture to add fine details of the image. Fig. 16 shows a visualization of the VAE features. We used a set of 20 generated face images and extracted features from different decoder layers of the UNet and VAE decoders, at the last time step (t=0) similar to that of PNP Tumanyan et al. (2023). We observed that the use of VAE decoder resulted in higher-frequency details than the UNet decoder. While the features from the UNet decoder contain semantic information, the features from the VAE decoder produced finer details such as hair, wrinkles, and small letters.

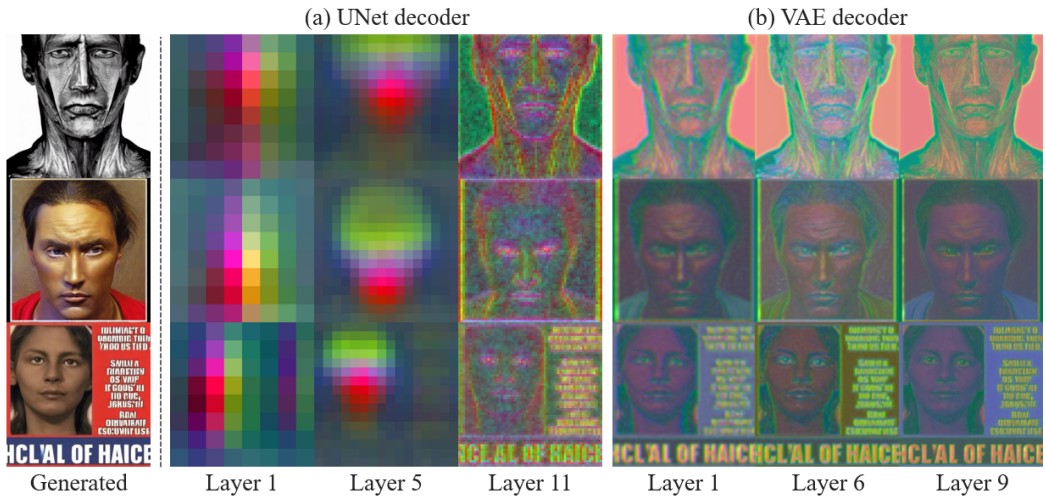

Figure 16: Extended visualization of features from UNet and VAE. (a) shows the UNet decoder features in lower resolution (layers 1), intermediate resolution (layers 5), and higher resolution (layers 11). (b) shows the VAE decoder features in lower resolution (layers 1), intermediate resolution (layers 6), and higher resolution (layers 9).

## D. Condition Diffusion Sampling for Training

### D.1 Additional Details on CDST

As stated in the main paper, we randomly sampled 100k prompts to estimate the distribution of SD. Specifically, we tokenized and embedded these 100k prompts in the space of the CLIP model. With this embedding, we conducted PCA to extract 512 principal components. We then checked the normality of the sampled embeddings with all 512 principal component axes using the Shapiro-Wilk test Shapiro & Wilk (1965) with a significance level of $\alpha = 5\%$.

As a result, 214 components rejected the null hypothesis of normality. This indicates that each of its marginals cannot be assumed to be univariate normal. Next, we conducted the Mardia test Mardia (1970; 1974) with the same 100k samples, taking into account skewness and kurtosis to check if the distribution is multivariate. The results failed to reject the null hypothesis of normality with a significance level of $\alpha = 5\%$. Therefore, we assumed $D_{SD}$ as a multivariate normal distribution for our sampling during training. In addition, we calculated the Earth Moving Distance (EMD) Levina & Bickel (2001) with 100k samples from LAION- 400M, which were not used for our analysis. For comparison, we used the normal distribution for each axis, and the uniform distribution to find that our $\mathcal{N}$ (244.22) is lower than the normal distribution for each axis (244.31) and the uniform distribution (1480.57).

## E. Additional experiments

### E.1 Additional Comparison with Baselines

As stated in the main paper, we presented four additional methods that failed to extract sketches following the desired style or exhibited severe artifacts. As shown in Figure 17, few-shot finetuning methods Ruiz et al. (2023); Gal et al. (2022) were unable to extract sketches when trained with a single example. The results of ControlNet Zhang et al. (2023b) showed severe artifacts because the method was originally proposed to be trained with thousands of images. Diffstyle Chung et al. (2024a), on the other hand, failed to preserve the content of source image. We also calculated LPIPS, SSIM, and FID scores, as in our main experiments, and as noted in Tab. 6, our method achieved the highest scores across all metrics.

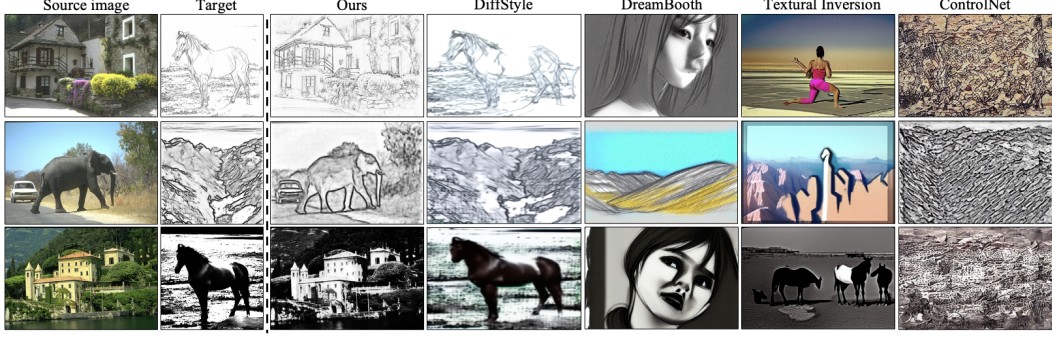

Figure 17: Experiment results on comparison with four additional baselines.

### E.2 Examples in Experiments

We presented quantitative results and visual comparison with and without using CDST for the ablation study described in the main paper. Here, we visualize additional results of the study in Fig. 18. For a perceptual study, a total of 23 participants were asked to make 20 different comparisons and determine which sketch style appeared most similar to the target sketch. Examples of our perceptual study is provided in Fig. 19 and Fig. 20.

Table 6: Quantitative comparison of different methods on the BSDS500 datasets. Best scores are denoted in bold, and the second-best are underlined.

| Methods | BSDS500 - anime | | | BSDS500 - HED | | | BSDS500 - XDoG | | | BSDS500 - average | | |
|---|---|---|---|---|---|---|---|---|---|---|---|---|
| | LPIPS | SSIM | FID | LPIPS | SSIM | FID | LPIPS | SSIM | FID | LPIPS | SSIM | FID |
| Ours | **0.218** | **0.493** | **126.5** | **0.227** | **0.593** | **110.6** | **0.143** | **0.649** | **62.8** | **0.196** | **0.578** | **100.0** |
| DiffStyle | 0.542 | 0.361 | 206.7 | 0.572 | 0.124 | 422.2 | 0.676 | 0.069 | 317.6 | 0.597 | 0.185 | 315.5 |
| DreamBooth | 0.806 | 0.302 | 233.5 | 0.746 | 0.185 | 277.8 | 0.723 | 0.195 | 276.1 | 0.758 | 0.227 | 262.5 |
| TI | 0.828 | 0.264 | 284.2 | 0.771 | 0.164 | 313.1 | 0.647 | 0.220 | 237.4 | 0.749 | 0.216 | 278.2 |
| ControlNet | 0.901 | 0.021 | 303.3 | 0.699 | 0.028 | 328.7 | 0.627 | 0.031 | 278.7 | 0.742 | 0.027 | 303.6 |

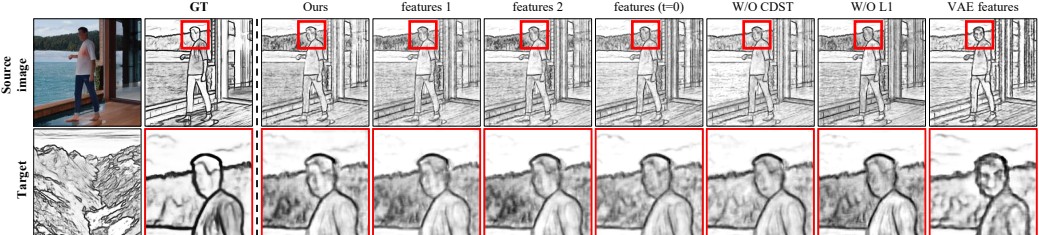

Figure 18: Visual examples of the ablation study. Ours generates higher quality results with details such as face, separated with hair region, compared to the alternatives.

## E.3 QUALITATIVE RESULTS

We present additional results of our method extracted in diverse styles which share the source image, in Fig. 21 and those of the comparison with baselines in Fig. 22. The additional comparison results further confirm that DiffSketch$_{distilled}$ extract superior results compared to the baseline methods.

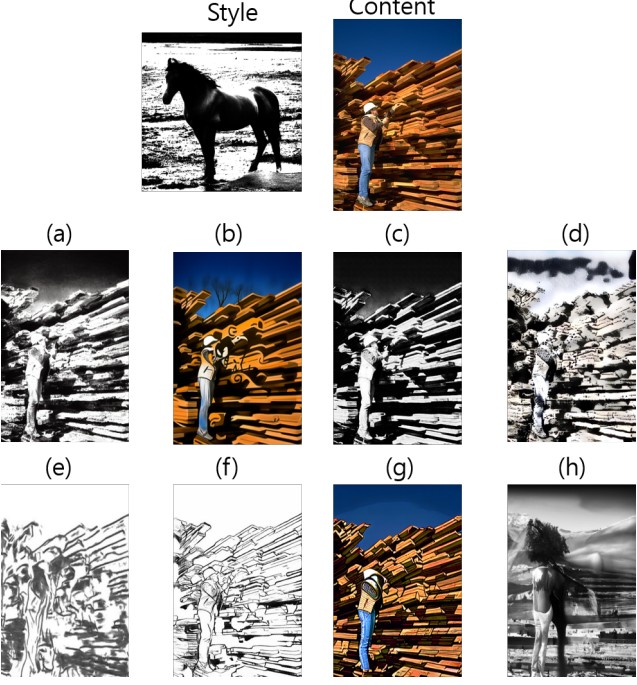

Figure 19: Example results of perceptual study. Participants were asked to choose one sketch image that has a style most similar to the style image while preserving the content of the content image faithfully. (c) is ours.

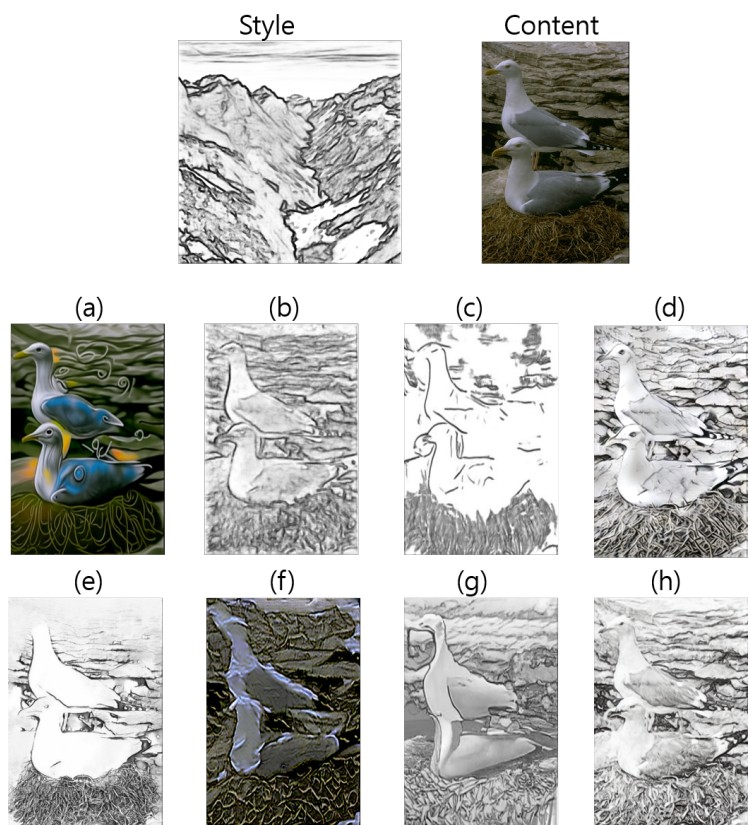

Figure 20: Example results of perceptual study. Participants were asked to choose one sketch image that has a style most similar to the style image while preserving the content of the content image faithfully. (b) is ours.

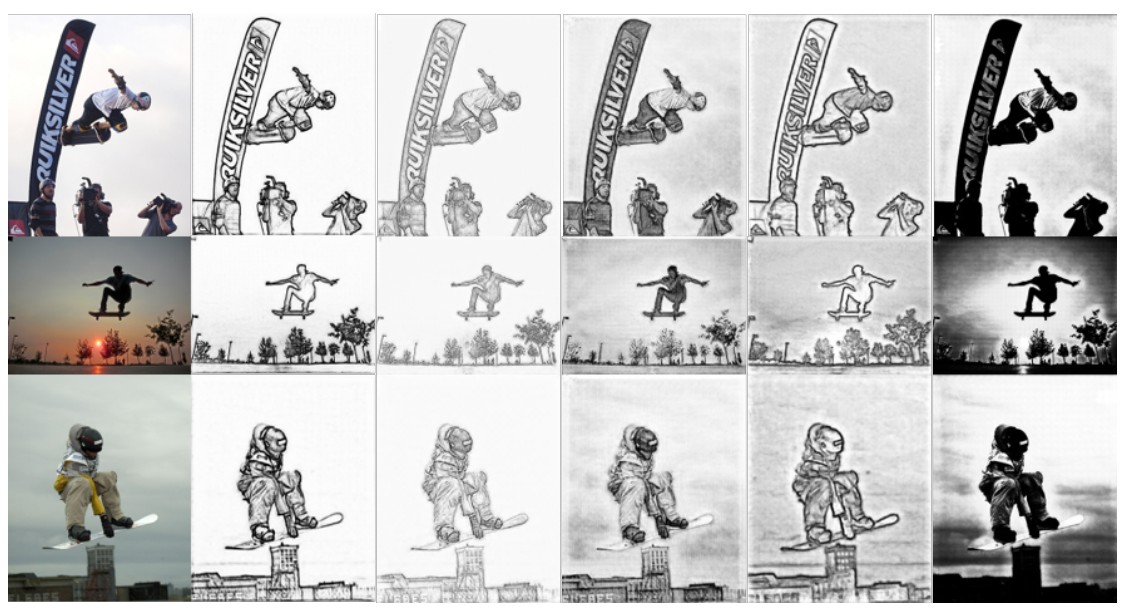

Figure 21: Additional results of Diffsketch$_{distilled}$ from shared sources.

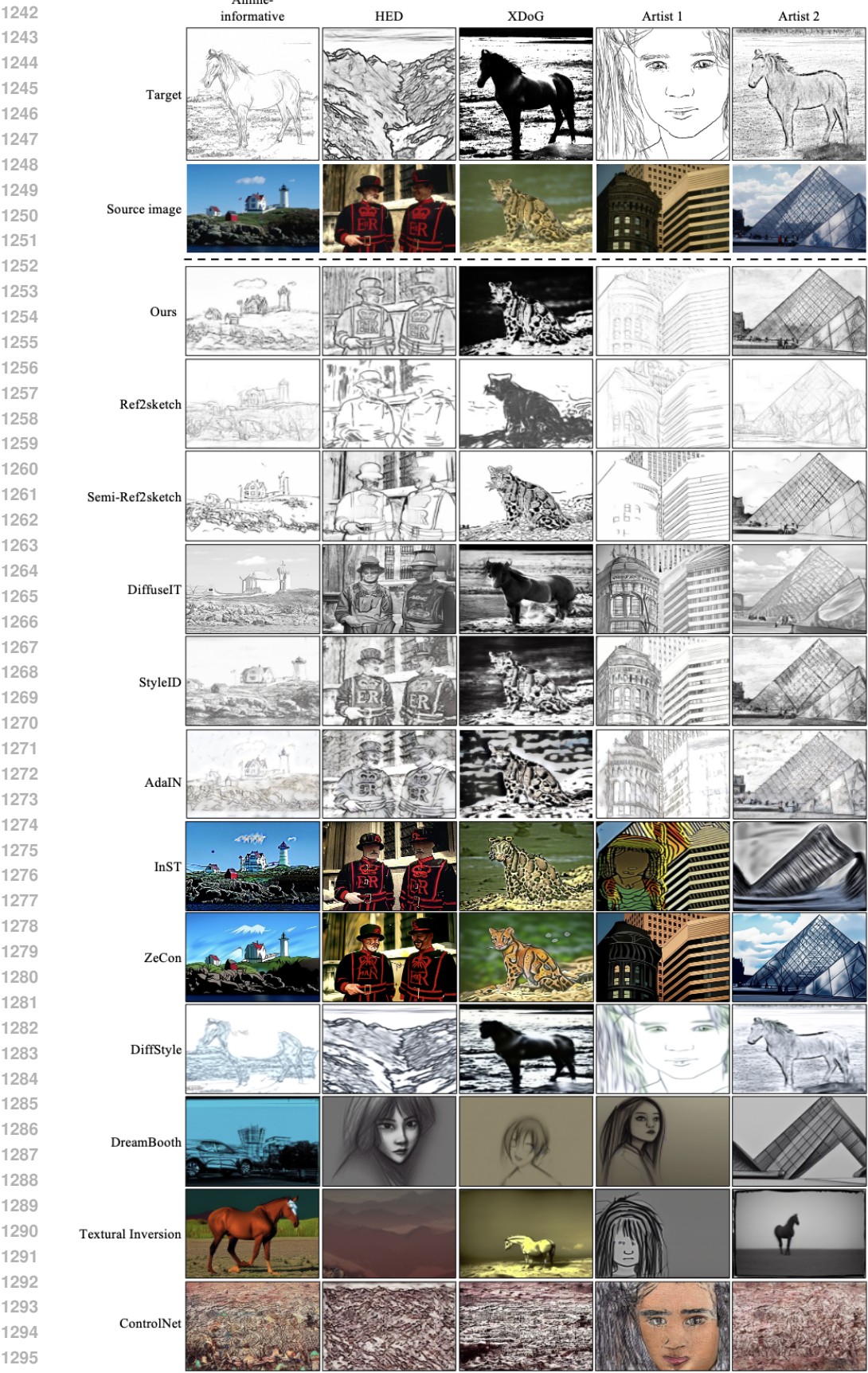

Figure 22: Qualitative comparison with alternative sketch extraction methods on the BSDS500 dataset.

