# OpenReview forum: "Stable Diffusion Feature Extraction for Sketching with One Example"
_ICLR.cc/2025/Conference — ICLR 2025 Conference Withdrawn Submission_

### Official Review · Reviewer_k2xY · 2024-10-29

**Soundness:** 3
**Presentation:** 2
**Contribution:** 2
**Rating:** 5
**Confidence:** 4

**Summary:**

This paper proposes a sketch extraction method with only one style example. The main idea of this paper is to train a sketch generator to predict the sketch images from the photo images generated by a fixed pre-trained diffusion model. The sketch generator integrates features from the diffusion model and the VAE decoder, and is trained on one image-sketch pair through the CLIP-based directional loss. After training, a pix2pix-based framework is trained to distill the abundant paired data generated by the diffusion model and the sketch generator for fast inference. The contributions lie in the idea to use diffusion models to generate paired data to solve the one-shot training problem with a special sampling strategy to ensure the diversity of the generated data.

**Strengths:**

**Originality** The main idea of analyzing the diffusion features to select and aggregate valid features makes sound to me. In addition, the proposed diffusion-based sampling scheme to generate diverse examples is interesting to me.

**Weaknesses:**

**Poor presentation**. The details of this paper are generally hard to follow. This paper contains many submodules and process. At least, a summarized algorithm could help the reader to understand the full process.

I found this paper is not self-contained. Many parts need to refer to the Appendix to help understand. See [Questions] for the details.

The reference format is poor. All the reference uses \citet, making it difficult to tell the main text and reference apart. Should use \citep instead! (`large datasets Seo et al. (2023)` -> ` large datasets (Seo et al. 2023)`.)

**Limited applications** This paper claims that the `method is tailored specifically for sketch generation`. However, I didn’t see any designs that only work for sketches. XDoG looks just like the binarized image rather than sketch image. And if the user draws a stylish image rather than a sketch of the input image, this method can still train. In the original paper of CLIP-based directional loss, the StyleGAN can be trained for various style editing tasks in addition to the sketch style. This paper only shows applications on sketches, which is limited.

In addition, the authors use HED, XDoG as two style types. These two types of sketch extraction are known, which has little value to invest how to imitate the sketch style. Why we have to train such complicated pipeline to imitate the simple HED and XDoG sketches? More complicated human-drawn sketches are what we truly want.

**Questions:**

Questions on unclear details and poor presentations:

1.	Line 173, `Fig 2` should be `Fig. 2`
2.	What does the feature gate $G*$ mean?
3.	In Fig. 2, there are 12 curves. Why there are 12 curves? 12 corresponding to what is not very clear to me.
4.	Eq. (2). There is no definition of $v_{i,n}$.
5.	Line 236, $CH$ should be $\text{CH}$
6.	Figure 4, $U_{md}$ should be $U_m$
7.	Line 313, what prompt C is used?
8.	Line 293, `$I_{source}$ and $I_sketch$` should be `$I_sketch$ and $I_{source}$`
9.	Line 325, avoid using $S$ since $S$ has been used in Eq. (6) and has different meanings
10.	Line 338 and Line 350, the regularization is not given. How to employ regularization?
11.	Line 354, why not using more test data to perform FID evaluation?

About experimental results

1.	Please provide the scores of Equal Feature in Table 2
2.	The authors show good performance on HED and XDoG, but w/o CDST has better performance on anim. However, HED and XDoG are less similar to the real human-drawn sketch styles. While anim looks more like human-drawn sketches. Does this mean the propose CDST is not suitable for the human-drawn sketches?
3.	Figure 7, please include the real human-drawn sketches for visual comparison.
4.	The results in the supp. such as Figure 22 show that the proposed method fails to imitate the Artist 1’s style as Semi-Ref2sketch. The proposed method fails to generate clean and sparse sketches. Please explain this limitation.

---

### Official Review · Reviewer_tzm8 · 2024-11-01

**Soundness:** 2
**Presentation:** 2
**Contribution:** 2
**Rating:** 5
**Confidence:** 3

**Summary:**

This paper proposes a diffusion based method to convert images to line drawings. The style of desired line drawing can be specified by a reference image

**Strengths:**

The high-level insight seems reasonable – manipulating the distribution of network features during diffusion process is a reasonable choice for achieving this sketching visual effect.

Supplemental material with both quantitative and qualitative data.

**Weaknesses:**

Although the high-level insight seems okay, the details of the method is extremely difficult to understand for me. I spent one afternoon to understand section 3.1 and 3.2 and still have no idea how this works. The “feature selection” and “aggregation” somewhat also links to “Open vocabulary panoptic segmentation with text-to-image diffusion models” and “Diffusion hyper features: Searching through time and space for semantic correspondence” but those previous “aggregation” are some sparse point or sparse mask for diffusion features. To me these does not explain what is the idea behind the stylization.

The “sketch generator” in 4.1 seems a distilled model trained from stable diffusion. To me it seems the stylization comes from the training of the model on the reference, not from some “diffusion feature aggression”?

Also it is not clear why we need to modify the VAE. The results in this paper do not look difficult to process for any existing SD VAEs.

**Questions:**

The objective also involves CLIPsim, can this be studied with ablative?

Also the images in the PDF file is very compressed, making it difficult to evaluate the quality.

---

### Official Review · Reviewer_H2CG · 2024-11-03

**Soundness:** 2
**Presentation:** 2
**Contribution:** 2
**Rating:** 3
**Confidence:** 5

**Summary:**

The paper proposes an algorithm for on shot style transfer given an input image to a sketch style. It makes crucial information regarding pre-trained knowledge within Stable Diffusion and its biases and leverages them to their advantage. Further the paper addresses limitations in their proposed approach and proposes efficient techniques to overcome them, as in their novel sampling technique. Lastly the work compares with state of the art sketch based style transfer algorithms and show that the proposed algorithm provided substantial improvement.

**Strengths:**

[+] The paper makes novel observations regarding Stable Diffusion (SD) with proper justification for hyperparameter selection and makes efficient use of inherent bias within SD for one shot style transfer between sketch and images. Especially regarding (i) the choice of number of clusters, as well as the observations across different timesteps, (ii) the kind of features extracted by UNet and VAE decoder.

[+] The paper makes astute observations regarding the limitation of CLIP for sampling scheme and addresses them

**Weaknesses:**

[-] Ablative study section of this paper is very weak. It is missing ablation studies of the different losses used and provides only for the L1 loss. However, according to the claims made in the paper, all of the proposed losses are very important. Thus, it is necessary to quantitatively and qualitatively judge their contribution in the final output.

[-] Seeing the importance of L1 loss via ablation studies a hyperparameter search for weight of L1 (and other losses) seems crucial to make the most out of the proposed method.

[-] Readability is hindered by the quality of sentence constructions throughout the entire paper. The entire paper should be revisited for better English and sentence construction

[-] The different sections in the paper are organized very poorly and the reader often has to move multiple sections to understand working of a particular concept described within the paper.

[-]  One of my major concerns is that – it is not at all clear how the distillation is happening in Sec. 4.4 to get the "DiffSketch_{distilled}" model. It briefly says about Pix2PixHD model, without any kinds of detail on the distillation. This section is extremely vague.

[-]  In ablation study, "one timestep" gives competitive performance to the proposed method and as per [A], timestep has a huge impact on the performance of the model so including comparison with results at a range of steps would be useful in verifying the robustness of the model.

[-] Through as per Table 4 the proposed algorithm works well, the dataset used for validation is very small, combined with the algorithm requiring the user to draw a sketch severely limits the algorithm's capabilities and its adaption for current sketch based datasets.

[-] The major contribution of the paper is the feature combination of SD and VAE. It would have been great to see a quantitative comparison of SD+VAE and SD only feature extraction.

Reference:
A. Denoising Diffusion Implicit Models, ICLR 2021.

**Questions:**

See weaknesses

---

### Official Review · Reviewer_rcCX · 2024-11-03

**Soundness:** 2
**Presentation:** 2
**Contribution:** 1
**Rating:** 3
**Confidence:** 4

**Summary:**

This article focuses on the technology of image or text-driven sketch extraction and generation based on one example. The research proposes a feature selector that can accurately screen the most discriminative features from the SD model. Subsequently, through a carefully designed feature aggregator, the organic integration of multi-level features is achieved. On this basis, a feature decoder is used to generate the corresponding sketches. The article further delves into the impact of features at different timesteps on the sketch generation process and innovatively proposes a set of new evaluation criteria, providing strong theoretical support for research in the field of sketch generation.

**Strengths:**

In the exploration of the extraction or generation of sketches from images or text, researchers often face the challenge of insufficient paired data (sketch-text pair or sketch-image pair). This paper ingeniously utilizes the existing text2img generation models, effectively reducing the dependence on large-scale datasets and achieving the capability of generating sketches with just a single sample. In order to more accurately evaluate the effectiveness of the generated sketches, this paper proposes a new set of evaluation criteria. Moreover, the paper conducts a comparative analysis with many existing methods. Through extensive experimental validation, the method proposed in this paper demonstrates its superiority and efficiency in multiple aspects.

**Weaknesses:**

This paper slightly lacks in terms of technological innovation and fails to contribute new perspectives or insights to the field of sketch generation. Additionally, the experimental design in the paper seems to lack the persuasive power to fully demonstrate its arguments. Although concepts such as "personalized sketch extraction" and "sketch style" are mentioned in the text, the experiments do not delve into the deep exploration of these areas. Furthermore, the paper seems to be somewhat confused in distinguishing between boundaries extracted from images and hand-drawn sketches, failing to make a clear distinction between the two. Finally, the logical structure of the article seems to require further refinement and optimization.

**Questions:**

i) boundaries or edgemaps can be extracted from images, but sketches can only be hand-drawn or generated, not extracted from images.
ii)In this paper, the definitions of "personalized sketch extraction" and "sketch style" require further clarification. The article treats various contour extraction techniques as different styles, which, however, is significantly different from the true concept of individual style.
iii) The BSDS500 dataset is meticulously constructed for edge detection and does not include any sketches. Although the edges are carefully annotated boundaries collected from multiple users, there remains a significant difference when compared to hand-drawn sketches. Hand-drawn sketches are characterized by their unique abstraction and morphological variations, setting them apart from precise edge annotations, which poses one of the main challenges in the field of image-to-sketch generation (image2sketch). Therefore, how do the experimental results on the BSDS500 dataset demonstrate the superior performance of the proposed method in the realm of sketch generation?
iv)The paper claims to possess the ability akin to one-shot learning, but the specific details of this capability do not seem to be clearly articulated within the text.

---

### Official Review · Reviewer_nfAq · 2024-11-04

**Soundness:** 3
**Presentation:** 2
**Contribution:** 2
**Rating:** 5
**Confidence:** 4

**Summary:**

The paper proposes a novel method called DiffSketch for generating sketch-style images from natural images based on a reference sketch style. The key innovation lies in utilizing features from a pretrained Stable Diffusion model to perform sketch generation with only one example sketch for training, addressing the challenge of data scarcity in sketch datasets. The method involves selecting representative features from multiple timesteps of the diffusion process and aggregating them to train a sketch generator that can generalize to various images. Additionally, the authors introduce a distillation process to streamline the model for efficient inference.

**Strengths:**

1. The paper introduces a novel method of using features from a pretrained Stable Diffusion model for sketch generation, which is a fresh perspective for this task.
2. By requiring only one reference sketch for training, the proposed method addresses the common issue of limited sketch datasets.
3. The authors provide thorough analysis and justification for their feature selection and aggregation process.

**Weaknesses:**

1. The target sketches used in this work are not real human-drawn sketches, and the resulting sketches differ significantly from those drawn by humans. This raises questions about the applicability of the method to authentic sketch generation.
2. The experiments primarily compare with style transfer works and a few sketch extraction methods, lacking comparison with relevant works like DiffSketcher, CLIPasso, and Clipascene.
3. The evaluation is conducted on edge extraction datasets, which may not fully represent the diversity of real-world sketches. Testing on datasets with real human sketches, such as TU-Berlin or Sketchy datasets, could provide a more comprehensive assessment.

**Questions:**

I wonder the performance of this method on some real sketch dataset, such as Sketchy dataset.

[1] The Sketchy Database: Learning to Retrieve Badly Drawn Bunnies, TOG 2016.

---

### Official Review · Reviewer_cEyR · 2024-11-05

**Soundness:** 2
**Presentation:** 2
**Contribution:** 2
**Rating:** 5
**Confidence:** 3

**Summary:**

This paper introduces DiffSketch, a novel method for generating sketches from text or images, using only a single drawing example for training.
1.	The proposed method explores the features of various layers and timesteps from a pretrained stable diffusion model. The proposed sketch generator aggregates the selected features for the SD model and a pretrained VAE decoder and generates a pair of image and sketch.
2.	To train the sketch generator G_sketch, a triplet, consisting of the diffusion feature, a generated image, and a manually drawn sketch for the image, is required. The training loss follows the definition of Mind-the-gap [Zhu et al 2022]. A novel sampling scheme, condition diffusion sampling for training (CDST), is proposed to ensure the diversity of training samples.
3.	While training the G_sketch from a single pair of generated image and drawn sketch requires high computation and memory cost, this work further trains a distilled version, DiffSketch_distilled, using the image-to-image translation framework with 30k generated pairs generated using DiffSketch.

**Strengths:**

The proposed two-level aggregation (SD features+VAE) makes full use of SD UNet features and VAE features to capture both overall structure and high-frequence details in generating high-quality images.

**Weaknesses:**

1.	The framework is not flexible for practical use. A manually drawn sketch is required for a generated image with the diffusion features to train the sketch generator. However, this is not easy to obtain. In the experiments, the authors use three sketch styles that can be automatically generated for quantitative evaluation. However, existing sketch pairs cannot be used for training.
2.	The ablation study shows that the two-level aggregation (SD features+VAE) and L1 loss are the most effective designs. The proposed CDST and SD feature selection bring weak improvement.
3.	This paper should compare with the sota Style Injection in Diffusion works [Chuang-CVPR 2024]. BTW, the work “ Jiwoo Chung, Sangeek Hyun, and Jae-Pil Heo. Style injection in diffusion: A training-free approach for adapting large-scale diffusion models for style transfer. In Proceedings of the IEEE/CVF Conference on Computer Vision and Pattern Recognition, pp. 8795–8805, 2024a.” is cited twice (same as Chuang-2024b). In the supplementary results, the authors only compare with DiffStyle, not the results from Chuang-CVPR 2024.
4.	The organization and writing of this paper could be improved in many ways. For example, the section organization is confusing. For example, Sec. 3 and Sec. 4 could be reorganized since Sec. 3.2 and 3.3 describe the detailed process of G_sketch and are not related to Sec. 3.1. Sec. 4.1 is the same as Sec. 3.2 and 3.3.

**Questions:**

1)	In the feature selection stage, is the clustering performed for every training image from all its LxT features or clustering from all training images? While there should be K cluster centers, what is the definition of the feature gate G*?
2)	Why set lt=10 in Eq 1?
3)	What is the difference between the first PCA figure and the second one in Fig 2? Features of two training images?
4)	In Fig. 6, it seems that every sketch contains a person/character in the bottom row, showing different content from the top row. Without showing the corresponding reference image, it is not clear how the content of the input source image is preserved.
5)	In Table 2 and Table 4, what model is used? DiffSketch or the DistilledDiffSketch?
6)	The training time and inference time are not clear to me. The training time of DiffSketch is about 3 hours by sampling 1000 times in CDST. The average inference time for DiffSketch is 4.74s. What is the input of DiffSketch during inference? Based on my understanding, DiffSketch requires the SD features to generate a pair of image and sketch. It cannot directly transfer an image to a desired sketch style.
7)	What is the distance in Table 3? Distance between which features? For example, each image has 13 feature cluster centers and what are the distances? Distance between 13 cluster centers and all features? If so, the Euclidean distance is definitely smaller than random sampling or equal-time sampling. This distance does not present much information.
8)	When compared with other methods, which model is used? If DiffSketch_distilled is used for comparison, 30k image-sketch pairs are required to train this model. It takes 4.74 seconds to generate each pair using DiffSketch, so it takes about 150k seconds (50 hours) to generate 30k sketch pairs to train DiffSketch_distilled. It is not fair to just say the proposed method makes inferences in 0.014s with just a single training example in Fig. 1.
9)	Some works have been officially published. For example, Luo’s work Diffusion hyperfeatures: Searching through time and space for semantic correspondence in Neurips 2023.

---

### Note · Authors · 2024-11-20

**Comment:**

Thank you to the reviewers for their valuable comments and feedback. We have decided to withdraw our submission and sincerely appreciate your time and effort in reviewing our work.

**Withdrawal Confirmation:**

I have read and agree with the venue's withdrawal policy on behalf of myself and my co-authors.